# Evaluation of Anthocyanin Profiles in Various Blackcurrant Cultivars over a Three-Year Period Using a Fast HPLC-DAD Method

**DOI:** 10.3390/foods10081745

**Published:** 2021-07-29

**Authors:** Barbora Šimerdová, Michaela Bobríková, Ivona Lhotská, Jiří Kaplan, Alena Křenová, Dalibor Šatínský

**Affiliations:** 1The Department of Analytical Chemistry, Faculty of Pharmacy in Hradec Králové, Charles University, 500 05 Hradec Králové, Czech Republic; barbora.simerdova@gmail.com (B.Š.); misa.bobrikova@gmail.com (M.B.); lhotski@faf.cuni.cz (I.L.); 2The Research and Breeding Institute of Pomology Holovousy, Holovousy 129, 508 01 Hořice, Czech Republic; j.kaplan@centrum.cz (J.K.); alenakrenova189@gmail.com (A.K.)

**Keywords:** blackcurrant, anthocyanins, delphinidin-3-rutinoside, cyanidin-3-rutinoside, delphinidin-3-glucoside, cyanidin-3-glucoside, HPLC, core-shell column

## Abstract

Anthocyanins are the most important polyphenolic substances contained in blackcurrant fruits. They are responsible for the various health benefits caused, in particular, by their high antioxidant activity. Anthocyanins derived from anthocyanidins cyanidin and delphinidin are typical for blackcurrant fruits, especially their rutinoside and glucoside forms. These four anthocyanins usually represent about 97–98% of total anthocyanins in blackcurrant fruits. In this study, we developed and validated a new HPLC-DAD method for rapid anthocyanin separation and determination in fifteen perspective blackcurrant cultivars (‘Ruben’, ‘Ben Lomond’, ‘Ben Conan’, ‘Ceres’, ‘Moravia’, ‘Ometa’, ‘Lota’, ‘Fokus’, ‘Tenah’, ‘Sejanec’, ‘Consort’, ‘Triton’, ‘Ben Hope’, ‘Ben Gairn’, and one gooseberry hybrid ‘Josta’). Eight of them were monitored throughout the three-year experiment. The most represented anthocyanins in all monitored blackcurrant cultivars were delphinidin-3-rutinoside (36.7–63.6%), cyanidin-3-rutinoside (26.4–40.6%), delphinidin-3-glucoside (6.1–17.9%), and cyanidin-3-glucoside (1.3–9.9%). The individual anthocyanin proportion (%) in each cultivar was specific, and a similar profile was verified in a three-year period for eight available cultivars. Total anthocyanin content expressed as a sum of four major anthocyanins present in blackcurrants was compared with values expressed as the equivalent of cyanidin-3-glucoside, as many authors do. We revealed an underestimation of about 20% with the latter method. Cultivars with the highest average total anthocyanin content were ‘Ben Gairn’ (294.38 mg/100 g), ‘Ceres’ (281.31 mg/100 g), and ‘Ometa’ (269.09 mg/100 g).

## 1. Introduction

Blackcurrant (*Ribes nigrum* L.) is a deciduous shrub in the family *Grossulariaceae* grown for its edible berries. Blackcurrants are the most important currants due to the large area of cultivation, high yields, and potential health benefits [1]. Blackcurrant berries quality is usually characterized by parameters such as taste, size, and appearance, and it is also one of the most examined species with respect to vitamin C and polyphenolic compounds [2,3]. Dominant blackcurrant phenolic compounds are anthocyanins; other notable phenolic substances are phenolic acids, flavonols, and proanthocyanidins [4].

Anthocyanins are natural water-soluble pigments responsible for the color of flowers and some fruits and vegetables [5,6,7,8].

Anthocyanins present in these fruits have been considered responsible for the various health benefits with disease preventive effects. Blackcurrants were labeled as ‘superfruit’ [3,4] for their health benefits. Significant antioxidant activity of anthocyanins plays an important role in the prevention of some diseases, particularly cardiovascular and nervous diseases, cancer, and diabetes [6,9,10]. Blackcurrant anthocyanins also have phytoestrogenic activity [10]. Therefore, their application potential is high, especially in the food industry, pharmacy, and cosmetic areas [6,9].

The concentration of total anthocyanins content in blackcurrant fruits usually varies between 80–470 mg/100 g of fresh weight (FW) [1,4,11]. It was found that blackcurrant berries may contain up to 15 anthocyanin structures [12]. However, only four anthocyanins are mainly present in blackcurrant. Anthocyanins derived from anthocyanidins cyanidin and delphinidin are typical for blackcurrant fruits, especially their rutinoside and glucoside forms (Figure 1). These four anthocyanins usually represent about 97–98% of total anthocyanins in blackcurrant fruits [12,13]. Minor anthocyanins occurring in blackcurrant fruits are peonidin-3-rutinoside and malvidin-3-rutinoside [2], and many others in trace amounts [12]. In some cases, the number of present anthocyanins in fruit samples can also be higher than twenty, as in the case of some blueberry cultivars [14].

Generally, anthocyanins as polar molecules are soluble in polar solvents. Thus, the most common solvents used for their extraction are aqueous mixtures of ethanol, methanol, and acetone [4,9,13]. Some authors use multiple-step extraction [13] or extraction with sulphured water [15]. Anthocyanins are more stable at low pH, which must be controlled during the extraction [16]. They are usually extracted from plant materials using methanol that contains a small amount of weak acid (e.g., formic acid) [5,6,9,10,17].

Separation techniques, especially liquid chromatography [9,12,18] and marginally as well capillary electrophoresis [6,19], are used for the quality and quantity anthocyanin evaluation. Historically, gas chromatography [20,21] and pH differential absorbance method [22,23] were applied for total anthocyanin determination. However, the latter did not enable the separation and identification of individual anthocyanins. It is a simple, quick, and economical method to determine only the total anthocyanin content [23]. Nowadays, the separation and quantification of anthocyanins are mainly achieved using reversed phase HPLC coupled with a PDA detector or mass spectrometry with acetonitrile and acidified water as the mobile phase [1,7,17].

Total and individual anthocyanins content are affected by genotype, harvest maturity of the fruits [18,24,25], and environmental factors [10,11,26]. Anthocyanins stability can also be affected by temperature and pH [7,27].

Few previously reported works were focused on anthocyanin profile in different blackcurrant cultivars [1,4,18,26,28], but, to our knowledge, no work dealt with individual anthocyanins analysis in more than five blackcurrant cultivars as an extended experiment lasting minimally three years. Mattila et al. [1] provided information about anthocyanins in 32 blackcurrant cultivars in dried fruits but only in one-year study. Milivojevic et al. [28] observed anthocyanins content of four cultivars in two years, and the aim was to investigate the effect of crop load on the accumulation and composition of anthocyanins and flavonols. Vagiri et al. [26] focused on variation in the genotype of three blackcurrant cultivars at two distant locations in Sweden during three years. The goal of the research study by Rubinskiene et al. [18] was to determine the composition and stability of five blackcurrant cultivars breeds at various ripeness phases. Nour et al. [4] identified nine anthocyanins present in three blackcurrant cultivars, Slimestad and Solheim [12] even reported and characterized fifteen anthocyanin structures based on a study with one cultivar ‘Titania’.

The objective of this study was to develop an efficient HPLC method using an advanced separation column with core–shell particle technology that allows the separation of blackcurrant anthocyanins in a short time. The method was applied to analyze four major anthocyanins, delphinidin-3-rutinoside (del-3-rut), cyanidin-3-rutinoside (cya-3-rut), delphinidin-3-glucoside (del-3-glu), and cyanidin-3-glucoside (cya-3-glu) in fifteen blackcurrant cultivars, their characteristic proportion in each cultivar, and estimation of minor anthocyanins fraction. The data coherence was revised during a three-year experiment for eight available cultivars.

## 2. Materials and Methods

### 2.1. Plant Material

Blackcurrant fruit samples (*Ribes nigrum*) were collected from the Research and Breeding Institute of Pomology Holovousy Ltd., City, Czech Republic. All cultivars were grown in one basically homogenous field. Berries of fourteen blackcurrant cultivars and one hybrid of blackcurrant and gooseberry were harvested from three bushes of each cultivar at the fully ripe stage. Cultivars included in our experiment were ‘Ruben’, ‘Ben Lomond’, ‘Ben Conan’, ‘Ceres’, ‘Moravia’, ‘Ometa’, ‘Lota’, ‘Fokus’, ‘Tenah’, ‘Sejanec’, ‘Consort’, ‘Triton’, ‘Ben Hope’, ‘Ben Gairn’, and hybrid ‘Josta’. Most of these cultivars are suitable for use as a table fruit and also characterized by high yield and good resistance to biotic and abiotic factors. The ‘Josta’ hybrid was the first bred cultivar of this type of crossing. It is the best known and also the most widespread blackcurrant hybrid in the world. Josta is characterized by large black aromatic fruits suitable for direct consumption.

Most of the selected blackcurrant cultivars were the same as in a previous study focused on the utilization of perspective blackcurrant cultivars as a table fruit [29]. Unfortunately, for important reasons, inappropriate harvest conditions and low yield, it was not possible to compare all 15 cultivars each year of the experiment. Therefore, just eight of these mentioned cultivars were successfully collected and measured each year of the experiment (‘Ruben’, ‘Ben Lomond’, ‘Ben Conan’, ‘Ceres’, ‘Moravia’, ‘Ometa’, ‘Lota’, ‘Fokus’). Samples were frozen immediately after harvesting and stored at −20 °C until processing.

### 2.2. Chemicals and Preparation of Solutions

Pure analytical standards of four major anthocyanins present in blackcurrant fruits were purchased from Extrasynthese (France). Namely, delphinidin-3-O-glucoside chloride, (purity ≥ 95%), delphinidin-3-O-rutinoside chloride (purity ≥ 95%), cyanidin-3-O-glucoside chloride, (purity ≥ 96%), and cyanidin-3-O-rutinoside chloride, (purity ≥ 96%). Methanol (gradient grade, Sigma Aldrich, Prague, Czech Republic), formic acid (anhydrous, Sigma Aldrich), and ultra-pure water were used. Ultra-pure water was purified through a Milli-Q system (Merck Millipore, Burlington, MA, USA).

Stock solutions of each standard were prepared individually in a concentration of 1 mg/mL in methanol. The mixed working solution at concentration 250 mg/L was prepared by mixing stock solutions aliquots. The working solution was used to prepare the calibration curve between concentration levels 10 and 250 mg/L. The desired concentrations were achieved by diluting the working solution with methanol. The whole calibration curve ranged between concentration levels 10 and 500 mg/L (*n* = 9).

The precision of the extraction procedure was carried out on eight sample solutions prepared as described in Section 2.3. Each of the blackcurrant berry solution was injected twice.

In short, berries extract for recovery evaluation were prepared in eight repetitions. The naturally present anthocyanin content was calculated using the calibration curve obtained with external standards in two of them (*c*1). The six other samples were spiked with a standard of known concentration 4 mg/L (*c*3) and then extracted by the same procedure as described below (Section 2.3), and the concentration was evaluated (*c*2). The recovery *R* (%) was calculated as the ratio:$$R\;\left(\%\right)=\frac{c2-c1}{c3}$$

### 2.3. Sample Preparation by Extraction

Anthocyanins are well soluble in polar solvents. Therefore, the choice of the extraction solvent was based on previously published works [5,6,9,10,17]. They are usually extracted from plant materials using methanol that contains small amounts of a weak acid (e.g., formic acid). Methanolic extracts were prepared according to the following method: frozen berries (about 100 g) were first thawed for few minutes and homogenized using a laboratory rod mixer (Sencor SHB 4460WH-EUE3, 1000 W). Two grams of the homogenate were weighed in triplicate for the extraction of anthocyanins with 10 mL of methanol containing 2% (*v*/*v*) formic acid. Extraction was done for twenty minutes in an ultrasonic water bath (Bandelin Sonorex RK 100, 35 KHz, 160 W, Germany). Extracts were filtered (0.45 µm PTFE filter) and immediately measured.

### 2.4. HPLC Analysis

The content of anthocyanins in blackcurrant samples was analyzed on a Shimadzu LC-10 HPLC system equipped with a diode array detector SPD—M10A VP, LC pumps 10 AC VP, a degasser DGU—14 A, an autosampler SIL—HTA, and a thermostat CTO—10 AC VP. The compounds were separated on an Ascentis Express C18, 150 × 4.6 mm, 2.7 μm particle size, analytical column (Supelco Analytical). The column was protected with a 5 × 4.6 mm security guard column. The column was operated at 50 °C with the elution solvents A (2% formic acid in water) and B (100% methanol) and a flow rate of 0.8 mL/min. The gradient was: 0–4 min, 5–20% B; 4–8 min, 20–25% B; 8–10 min, 25–90% B, 10–10.15 min, 90% B; 10.15–10.30 min, 90–5% B, and finally reconditioning of the column (10.30–13 min, 5% B). The injection volume was 10 μL of an extract and anthocyanins were detected at a wavelength of 520 nm. The total run time was 13 min. Four major anthocyanins were separated in a time of less than 10 min. Identification of anthocyanins in blackcurrant samples was achieved by comparing retention times and UV/Vis spectra with the retention times and spectra of the external standards. The quantity of each anthocyanin was calculated by the calibration curve of the external standards. Concentrations were expressed in mg per 100 g of fresh fruits. Each sample was prepared in three replications (3 × 2 g were weighed) for the elimination of unexpected influences during sample preparation.

## 3. Results and Discussion

### 3.1. HPLC Method Optimization

The aim of HPLC method optimization was to find the most suitable conditions for the determination of the four major anthocyanins occurring in blackcurrant berries, del-3-rut, cya-3-rut, del-3-glu, and cya-3-glu. The main requirement for the HPLC analysis was the efficient separation of four major anthocyanins and further minor anthocyanins with sufficient resolution and narrow peaks in a short time. A mixture of berries of few different cultivars of blackcurrant was used to optimize the chromatography separation conditions. Extraction of anthocyanins from homogenized raw material was carried out using methanol containing 2% formic acid. Extraction time was optimized for twenty minutes in an ultrasonic water bath at laboratory temperature.

The method selectivity for the separation of anthocyanins was very good due to their unique absorption properties at higher wavelengths as compared with other phenolic compounds. Absorbance spectra were observed, del-3-glu shows a maximum at 524 nm, del-3-rut at 526 nm, cya-3-glu at 516 nm, and cya-3-rut at 519 nm. The wavelength chosen for anthocyanins detection in our study was 520 nm, which is in line with other studies [9,18,23,30]. Other colored compounds visible at 520 nm were assumed to be minor undefined anthocyanins.

Conventional silica-based columns, most often the chemically modified phase with C-18 chains, are often used for anthocyanins separation. In this work, ten different advanced stationary phases were tested involving hybrid particles, superficially porous particles, and varying chemistries. The following columns Supelco C18, 250 × 4.6 mm, 5 µm particle size; Waters X-Bridge Phenyl, 150 × 4.6 mm, 3.5 µm particle size; Phenomenex Kinetex XB-C18, 150 × 4.6 mm, 5 µm particle size; Phenomenex Kinetex Biphenyl, 150 × 4.6 mm, 5 µm particle size; and Supelco Ascentis Express C18, 150 × 4.6 mm, 2.7 µm particle size, showed the best selectivity. The last column mentioned was chosen for use in our experiment based on the significant benefits of core–shell particle technology in anthocyanin separation, specifically increasing column efficiency and reducing analysis time. Methanol in combination with acidified water shows better separation properties than acetonitrile. Several different elution gradients were tested; various linear gradient slopes in combination with different concentrations of organic phase (from 2 to 10%) at the beginning of the gradient and various mobile phase flow rates (0.6 mL/min, 0.8 mL/min, and 1.0 mL/min) were included in the method optimization. The best results of peak symmetry and resolution in combination with the chosen column were obtained using 5% methanol at time 0, gradually increased to 90% in 10 min (described in Section 2.4). Different column thermostat temperatures (35 °C, 40 °C, 45 °C, and 50 °C) were tested. The highest temperature 50 °C provided a significant shortening of analysis time. Moreover, the higher temperature improved the peak symmetry of the separated anthocyanins. The developed method separates the major anthocyanins within 10 min. The total run time was 13 min when other undefined minor anthocyanins were eluted, and the column was equilibrated to initial conditions (Figure 2). Compared to the available literature, these times are considerably shorter; in articles, we often encounter data of about 30 min [2,9] or even 50 min [4,18].

### 3.2. Method Validation

After method optimization, the validation protocol was performed. Preparation of all samples for validation is described in Section 2.2. Linearity was measured for all four compounds at nine concentration levels 10, 20, 50, 75, 100, 150, 200, 250, and 500 mg/L. The correlation coefficient (R^2^) was consistently higher than 0.999. Peak resolution was in all cases higher than 2.0, and peak symmetry ranged between 1.01 and 1.13. The peak capacity factor for all major anthocyanins ranged between 3.18 to 4.27.

The injection repeatability of the HPLC method was tested in three concentration levels (20, 100, and 250 mg/L), and relative standard deviations of areas (RSD) were in all cases less than 1.0% (*n* = 7). The recovery values for the four major anthocyanins naturally occurring in blackcurrants (del-3-rut, cya-3-rut, del-3-glu, and cya-3-glu) were investigated. The result of recovery ranged between 90.10 to 92.63%, with precision RSD < 2.1%. The validation protocol confirmed the reliability of the developed HPLC-DAD method for the intended purpose. Validation data are presented in Table 1.

### 3.3. Evaluation of Individual and Total Anthocyanin Content in Blackcurrant Berries

The most represented anthocyanins in fourteen monitored blackcurrant cultivars were del-3-rut (36.7–63.6%), cya-3-rut (26.4–40.6%), del-3-glu (6.1–17.9%), and cya-3-glu (1.3–9.9%). These four anthocyanins usually represent more than 97% of total anthocyanins in blackcurrant fruits. This fact was confirmed in almost all cultivars that were tested, where minor anthocyanins ranged between 0.9 and 2.3%. In contrast, cultivar ‘Sejanec’ and ‘Tenah’ (Figure 2) showed higher representation, 7.2% and 4.8%, respectively, expressed as the sum of undefined peaks. According to available literature, unknown compounds, which are the minor part of blackcurrant fruits, are likely peonidin-3-rutinoside and malvidin-3-rutinoside [2].

The highest representation of del-3-rut was confirmed in almost all tested blackcurrants cultivars. Cultivars ‘Ceres’, ‘Ometa’, and ‘Sejanec’ had equivalent representation of both anthocyanidin rutinosides. Matilla et al. [1] observed 32 cultivars of blackcurrant during a one-year experiment leading to the conclusion that del-3-rut was the major anthocyanin in 25 cultivars, while a higher proportion of cya-3-rut was detected in six cultivars, which corresponds with our results. The highest proportion of del-3-rut was detected in cultivar ‘Triton’ (63.6%) and ‘Ben Gairn’ (55.7%). Cya-3-rut usually formed about 30–40%, in cases of ‘Triton’ (26.4%), ‘Ben Gairn’ (28.4%), and ‘Lota’ (29.0%) less than 30%. Del-3-glu had the highest ratio in cultivars ‘Ceres’ (17.9%) and ‘Ruben’ (16.9%); in other cultivars, it represented less than 16%, in the case of ‘Moravia’ and ‘Triton’ less than 7%. Cya-3-glu represented less than 10%.

Hybrid of blackcurrant and gooseberry, cultivar ‘Josta’, showed significantly different composition of anthocyanins, cya-3-rut was the most represented anthocyanin in this cultivar (60.7%), followed by del-3-rut (17.7%), cya-3-glu (12.9%), and del-3-glu (8.7%). ‘Josta’ cultivar contained only major observed anthocyanins.

Eight blackcurrant cultivars were monitored for all three years of the experiment (list in Section 2.1). Changes in representation (%) of individual anthocyanins contained in berries were recorded and compared. The overall summary and three-year anthocyanin profile of each cultivar are presented in Figure 3. The relative representation of delphinidin-3-rutinoside was stable in the year-on-year comparison. The relative standard deviation (RSD) of del-3-rut three-year proportion was less than 3% for most cultivars, RSD for cultivar ‘Ruben’ was 4.9%, and 8.3% for cultivar ‘Lota’. Cyanidin-3-rutinoside was comparatively stable during this three-year period except for the cultivars ‘Fokus’ (14.4%) and ‘Ben Lomond’ (7.6%). The proportion of both glucosides, which were not as high as rutinosides, was quite different in the year-on-year comparison for the same cultivars. RSD ranged between 2.2–39.3% for del-3-glu and 2.9–31.3% for cya-3-glu.

Many authors [1,15,30,31] express the total anthocyanin concentrations in fruit samples as an equivalent of cya-3-glu, which is the most widespread anthocyanin in nature. Rubinskiene et al. [18] expressed total anthocyanin content even to the equivalent of cya-3-rut, which is the second most abundant anthocyanin in blackcurrant. Our study found that quantification as cya-3-glu equivalents gave markedly lower results regarding the total anthocyanin content in comparison with the content expressed as a sum of individual anthocyanins, as summarized in Table 2. The estimation from cya-3-glu is misleading, underestimating the real concentrations of about 20% in comparison to the sum of four major anthocyanins. The most significant difference was found in the cultivar ‘Consort’ (26.3%), total anthocyanin content expressed as equivalent of cya-3-glu was 127.2 mg/100 g and total anthocyanin content expressed as the sum of individual anthocyanins was 172.6 mg/100 g. We concluded that using the only standard for quantification can cause underestimation of the total anthocyanin content due to the low representation of cya-3-glu in blackcurrant fruit. The conclusions of other research groups are in line with our statement [11,23]. Therefore, we expressed the anthocyanin content as a sum of major anthocyanins.

Complete values of total anthocyanin content throughout the three-year experiment are summarized in Figure 4 and in Appendix A. In the first year, the cultivar ‘Ometa’ clearly had the highest content 310.98 mg/100 g, and the second highest value had cultivar ‘Ceres’ with 261.82 mg/100 g. The significantly lowest content was found in cultivar ‘Ben Conan’ (79.41 mg/100 g). In the second year, the highest total anthocyanin content was again found in cultivar ‘Ometa’ (267.40 mg/100 g), with almost the same value, the second cultivar ‘Ruben’ (263.16 mg/100 g). In the second year of the experiment, no cultivar had a lower anthocyanin concentration than 100 mg/100 g. The highest results were found during the last year of the experiment. There were three cultivars that showed a higher value of total anthocyanin content than 300 mg/100 g, namely ‘Ben Gairn’ (353.59 mg/100 g), ‘Ceres’ (347.56 mg/100 g), and ‘Ben Hope’ (301.89 mg/100 g). Cultivar ‘Ometa’ contained only 228.91 mg/100 g in the third year of the experiment. The lowest total anthocyanin content in the last year was measured in hybrid ‘Josta’ (93.48 mg/100 g). There are markedly different results between the cultivars and also the year of the experiment. Crop variability according to climatic influences can be assumed. Rubinskiene et al. [18] also concluded that the concentration of anthocyanins in blackcurrant berries depends on genetic characteristics and ripeness. Significant differences among blackcurrant cultivars were detected in all observed anthocyanins by Milivojevic et al. [28]. Vagiri et al. focused on variation in genotype (three blackcurrant cultivars), location (two distant locations in Sweden), and year (three-year period). The changes have been demonstrated; regression analyses showed the effect of genotype to be significantly higher than that of location and year [26]. Anthocyanin content can be affected by the weather in a certain year.

Cultivars with the highest average total anthocyanin content (expressed as the sum of individual anthocyanins) were ‘Ben Gairn’ (294.38 mg/100 g), ‘Ceres’ (281.31 mg/100 g), and ‘Ometa’ (269.09 mg/100 g). These three cultivars were introduced into cultivation work at the turn of the new millennium. Their common denominator is earlier maturity and high yield potential. Other blackcurrant cultivars ranged between 154.74 mg/100 g and 256.55 mg/100 g; only hybrid ‘Josta’ had a significantly lower content (93.48 mg/100 g). The reported concentration of total anthocyanins content in blackcurrant fruits usually varies between 80–470 mg/100 g of fresh weight [1,4,11], which is in line with our results.

Chromatograms of several real samples are presented in Appendix A.

## 4. Conclusions

A new chromatography method for the fast separation of individual anthocyanins in blackcurrant cultivars has been developed. An Ascentis Express C18 150 × 4.6 mm, 2.7 µm, core shell particle chromatography column was chosen for fast and efficient separation. The developed method separated the major anthocyanins within 10 min while minor anthocyanins were eluted till the 13th minute of the analysis. Compared to the available literature, analysis times were considerably shorter than the reported 30 min [2,9] or even fifty minutes chromatography runs [4,18]. Based on the real sample investigation, we can conclude that there is a considerable difference in anthocyanin contents among the observed cultivars and also within the years in the same cultivars, as was reported elsewhere [32]. Interestingly, the proportion (%) of anthocyanins in individual cultivars is almost unchanged over three years and seems to be specific to them. Cyanidin and delphinidin rutinosides are dominating compounds. The four main anthocyanins represented more than 92% of total anthocyanins in all blackcurrant samples, mostly about 98%, which is in agreement with previous works [12,13]. We suggest estimation of the total anthocyanin content in blackcurrant as the sum of four major anthocyanins rather than the equivalent of cyanidin-3-glucoside, as many authors do, because cya-3-glu is the least represented anthocyanin in blackcurrants, and it leads to underestimation of about 20%. Cultivars with the highest average total anthocyanin content (expressed as the sum of individual anthocyanins) were ‘Ben Gairn’ (294.38 mg/100 g), ‘Ceres’ (281.31 mg/100 g), and ‘Ometa’ (269.09 mg/100 g).

## Figures and Tables

**Figure 1 foods-10-01745-f001:**
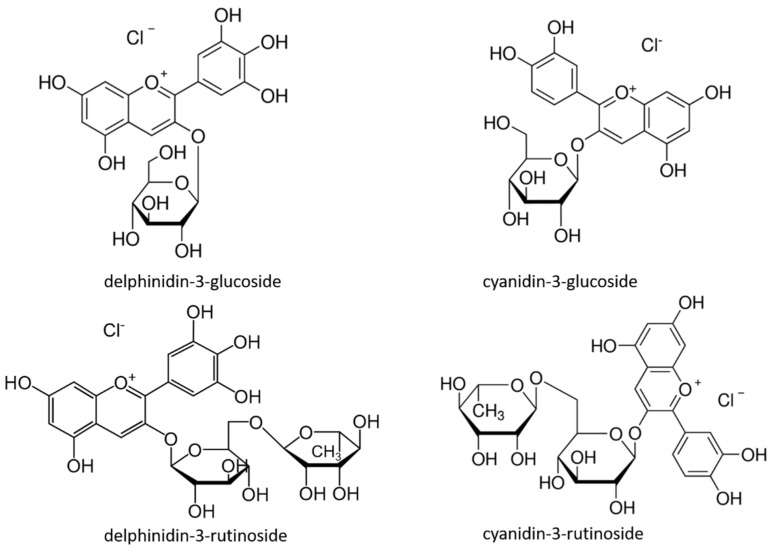
Chemical structure of the four major anthocyanins present in blackcurrant fruit.

**Figure 2 foods-10-01745-f002:**
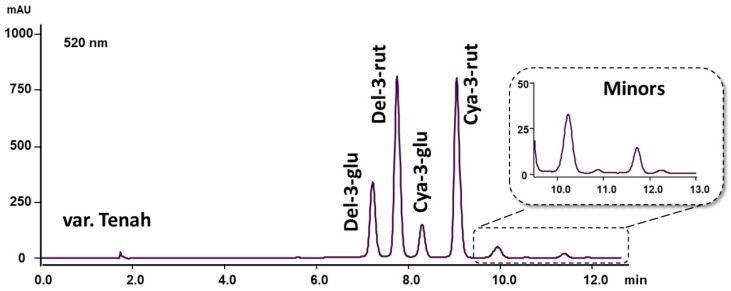
Chromatogram of major and minor anthocyanins in blackcurrant cultivar ‘Tenah’.

**Figure 3 foods-10-01745-f003:**
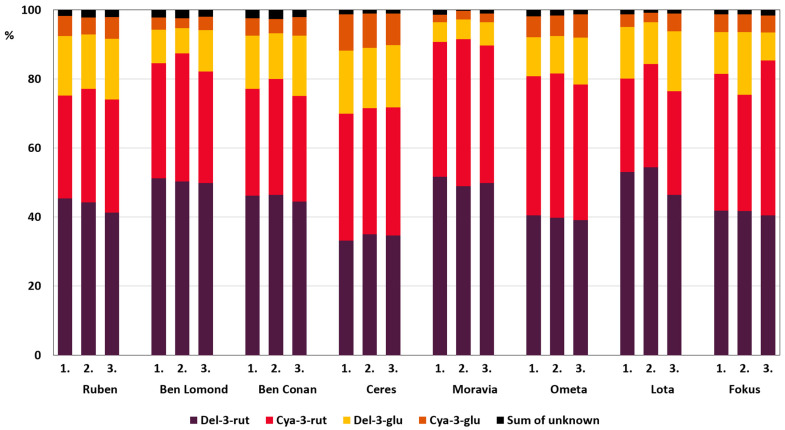
Representation (%) of blackcurrant anthocyanins in eight blackcurrant cultivars measured throughout the three-year experiment.

**Figure 4 foods-10-01745-f004:**
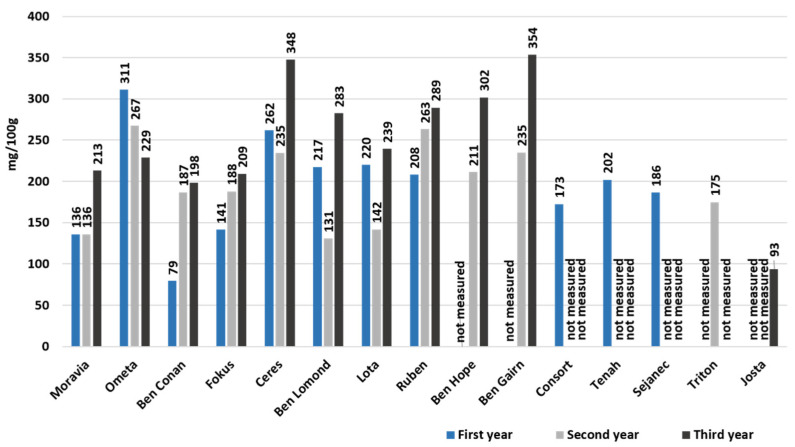
Total anthocyanin content (mg/100 g) in available blackcurrant cultivars measured during a three-year period (expressed as the sum of four major anthocyanins).

**Table 1 foods-10-01745-t001:** Validation parameters of the HPLC-DAD method for separation of four major anthocyanins.

Analyte	t_R_ (min)	Peak Resolution	Peak Symmetry ^1^	Capacity Factor P_c_ ^2^	Precision (RSD, %) ^3^	Recovery (%) ^4^
Delphinidin-3-glucoside	7.42	-	1.01	3.18	2.06	92.63
Delphinidin-3-rutinoside	7.92	2.09	1.04	3.50	1.81	90.10
Cyanidin-3-glucoside	8.56	2.06	1.13	3.81	1.40	91.07
Cyanidin-3-rutinoside	9.30	3.04	1.12	4.27	1.90	90.88

^1^ Peak symmetry was calculated by the Lab Solution chromatography software (ratio of descending to ascending part of the peak in 10% of height); ^2^ Peak capacity expressing efficiency of the method (gradient elution) is calculated as P_c_ = (the gradient time/4 × peak width in half) + 1; ^3^ Repetitive determination of blackcurrant sample (*n* = 8); ^4^ The method recovery was determined using six spiked blackcurrant samples at a concentration level of 4 mg/L.

**Table 2 foods-10-01745-t002:** Anthocyanin content (mg/100 g) in blackcurrant cultivars measured during the first year of the experiment—comparison of different evaluation approaches.

Cultivars	Total Anthocyanins (Cya-3-Glu Equivalent)	Sum of 4 Major Anthocyanins	Del-3-Rut	Cya-3-Rut	Del-3-Glu	Cya-3-Glu
Ometa	231.76	310.98	137.66	129.09	30.86	13.36
Ceres	212.15	261.82	97.08	101.71	42.22	20.82
Lota	171.55	219.95	125.92	60.73	28.60	4.70
Ben Lomond	169.58	217.38	120.32	73.68	18.95	4.42
Ruben	165.72	208.30	103.94	64.59	31.66	8.11
Tenah	150.67	202.02	84.49	78.57	29.13	9.83
Sejanec	169.21	186.27	74.88	72.29	26.46	12.63
Consort	127.19	172.56	89.49	58.28	20.06	4.74
Fokus	110.70	141.42	64.08	57.51	15.68	4.15
Moravia	103.66	135.78	74.30	53.10	7.80	0.58
Ben Conan	62.80	79.41	40.17	25.77	11.84	1.63

## Data Availability

Not applicable.

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
