# Peer review of "Evaluation of Anthocyanin Profiles in Various Blackcurrant Cultivars over a Three-Year Period Using a Fast HPLC-DAD Method"

_foods, 2021, doi:10.3390/foods10081745_

Round 1

Reviewer 1 Report

The paper is well-written and well-designed. The subject is important to be investigated, although some changes need to be done.

  • You should change the title as you were not able to determine the anthocyanin profile for the fifteen varieties of blackcurrant during the three years.
  • In Figure 1, you should use the same configuration for the sugar moiety. You should also verify the double bonds in the anthocyanin structure as I think you have some errors.
  • Lines 220-222: There is a repetition of this sentence three times in the manuscript. You should avoid repetition.

Author Response

Reviewer #1:

The paper is well-written and well-designed. The subject is important to be investigated, although some changes need to be done.

  • You should change the title as you were not able to determine the anthocyanin profile for the fifteen varieties of blackcurrant during the three years.

It is true that the title can be misleading to some readers. Therefore, we changed the title to "Evaluation of anthocyanin profiles in various blackcurrant cultivars over a three-years period using fast HPLC-DAD method "

The specification of all cultivars and the number of cultivars for three years study is reported in abstract.

  • In Figure 1, you should use the same configuration for the sugar moiety. You should also verify the double bonds in the anthocyanin structure as I think you have some errors.

It was corrected.

  • Lines 220-222: There is a repetition of this sentence three times in the manuscript. You should avoid repetition.

Based on this comment, the theoretical part concerning the extraction was removed from the chapter 3.1 HPLC method optimization (lines 220 – 223).

Reviewer 2 Report

The authors performed a very interesting study on the variability of the qualitative-quantitative profile of anthocyanins in selected black currant cultivars. However, the article requires several corrections to be suitable for presentation to the scientific community.

The title should be changed to not 15 but 8 cultivars, because in fact, 8 cultivars were monitored over a period of 3 years. Accordingly, this information should be changed in the abstract. Lines 18-21 in the abstract should be removed, which talk about the methodology. I also suggest removing verses 29-30, as not all research institutions can afford anthocyanin standards. Besides, there is a generally accepted standard procedure.

The introduction contains part of the methodology and many book descriptions concerning anthocyanins, but not referring to the scope of the study performed, therefore the introduction is too long. The following passages should be removed: lines 43-48, 55-65, 100-102, 124-126.

The fragment from verses 140-147 should be rewritten. Please do not mention previous studies if they have no substantive connection to the current one - it is confusing for the reader. The authors should write that they studied 15 varieties of black currant but for important reasons were able to study 8 varieties for 3 years in a row. Verse 148-149 should be moved behind verse 139.

Verse 169- from what I understood 8 samples were used for recovery studies but in 6 replicates. Verses 176-178 are a repetition of information so this section should be removed.

Please provide the model and manufacturer of the homogenizer mentioned in verse 186. Please provide the parameters of the ultrasonic bath (verse 188). Was the temperature controlled for the entire 20 minutes? What was the extraction temperature?

The description of the results and its discussion are clearly described. However, they also need improvement. Verses: 216-218, 220-224 are repetition of methodology therefore I recommend to remove these passages. In verse 224-225 the authors mention optimization of ultrasonic extraction time. Please provide more information - what times were tested and why (on what basis) an extraction time of 20 minutes was chosen?

In verse 243-244, the authors mentioned that they studied other gradients. Please state which ones and why the one used for the study was chosen.

The description of the results is in depth and there are comparisons with the literature. The main achievements of the research and conclusions are listed very concisely and specifically - I have no comments on them.

I recommend to include the following recently published items in the cited literature, which will greatly improve the manuscript:

  1. Impact of High-Pressure Homogenization Parameters on Physicochemical Characteristics, Bioactive Compounds Content, and Antioxidant Capacity of Blackcurrant Juice, 2021, https://doi.org/10.3390/molecules26061802
  2. The Effect of Organic and Conventional Cultivations on Antioxidants Content in Blackcurrant (Ribes nigrum L.) Species, https://doi.org/10.3390/app11115113

Author Response

Reviewer #2:

The authors performed a very interesting study on the variability of the qualitative-quantitative profile of anthocyanins in selected black currant cultivars. However, the article requires several corrections to be suitable for presentation to the scientific community.

  • The title should be changed to not 15 but 8 cultivars, because in fact, 8 cultivars were monitored over a period of 3 years. Accordingly, this information should be changed in the abstract. Lines 18-21 in the abstract should be removed, which talk about the methodology. I also suggest removing verses 29-30, as not all research institutions can afford anthocyanin standards. Besides, there is a generally accepted standard procedure.

Based on this comment (and also comment 1) of reviewer #1) we changed the title to " Evaluation of anthocyanin profiles in various blackcurrant cultivars over a three-years period using fast HPLC-DAD method ". The information in the abstract has been specified and changed on the basis of this recommendations.

  • The introduction contains part of the methodology and many book descriptions concerning anthocyanins, but not referring to the scope of the study performed, therefore the introduction is too long. The following passages should be removed: lines 43-48, 55-65, 100-102, 124-126.

According to the suggestion, we have shortened the introduction and removed recommended passages (lines 43-48, 55-65, 100-102, 124-126).

  • The fragment from verses 140-147 should be rewritten. Please do not mention previous studies if they have no substantive connection to the current one - it is confusing for the reader. The authors should write that they studied 15 varieties of black currant but for important reasons were able to study 8 varieties for 3 years in a row. Verse 148-149 should be moved behind verse 139.

The paragraph was modified as recommended.

  • Verse 169- from what I understood 8 samples were used for recovery studies but in 6 replicates. Verses 176-178 are a repetition of information so this section should be removed.

Lines 176-178 were removed.

  • Please provide the model and manufacturer of the homogenizer mentioned in verse 186. Please provide the parameters of the ultrasonic bath (verse 188). Was the temperature controlled for the entire 20 minutes? What was the extraction temperature?

Ultrasonic water bath  (Bandelin Sonorex RK 100, 35 KHz, 160 W, Germany) – added.

Laboratory rod mixer (Sencor SHB 4460WH-EUE3, 1 000 W) – added.

The temperature of the ultrasonic water bath was monitored throughout and corresponded to the usual laboratory temperature (max 25°C).

  • The description of the results and its discussion are clearly described. However, they also need improvement. Verses: 216-218, 220-224 are repetition of methodology therefore I recommend to remove these passages. In verse 224-225 the authors mention optimization of ultrasonic extraction time. Please provide more information - what times were tested and why (on what basis) an extraction time of 20 minutes was chosen?

Based on this comment, we have shortened this paragraph and removed recommended passages (lines 216-218, 220-224).

Extraction time of 20 min was optimized according the extraction yield. Time longer than 20 minutes did not increase the extraction yield of anthocyanins (evaluated as peak area).

  • In verse 243-244, the authors mentioned that they studied other gradients. Please state which ones and why the one used for the study was chosen.

The required information has been provided: …various linear gradient slopes in combination with different concentration of organic phase (from 2 to 10%) at the beginning of the gradient and various mobile phase flow rates (0.6 ml/min, 0.8 ml/min, and 1.0 ml/min) were included in the method optimization.

  • The description of the results is in depth and there are comparisons with the literature. The main achievements of the research and conclusions are listed very concisely and specifically - I have no comments on them.

Thank you for positive evaluation.

  • I recommend to include the following recently published items in the cited literature, which will greatly improve the manuscript:
  • Impact of High-Pressure Homogenization Parameters on Physicochemical Characteristics, Bioactive Compounds Content, and Antioxidant Capacity of Blackcurrant Juice, 2021, https://doi.org/10.3390/molecules26061802

In our manuscript, we do not focus on the processing of blackcurrants and blackcurrants products, so we will not include this article in the used sources. It is a different area of interest.

  • The Effect of Organic and Conventional Cultivations on Antioxidants Content in Blackcurrant (Ribes nigrum L.) Species, https://doi.org/10.3390/app11115113

The results of our study fit well with this report. Therefore, the reference was mentioned in conclusion and added.

Round 2

Reviewer 1 Report

The structure of delphinidin-3-glucoside in Figure 1 it is not correct.  The number of double bonds in the pyranic ring are wrong. You have one carbon that lacks a double bond. You should correct before publication.

Author Response

The structure of delphinidin-3-glucoside in Figure 1 it is not correct.  The number of double bonds in the pyranic ring are wrong. You have one carbon that lacks a double bond. You should correct before publication.

We believe that the structure of delphinidin-3-glucoside is correct. It is the same as in other sources:

  • Sigma Aldrich Delphinidin chloride analytical standard | 528-53-0 (sigmaaldrich.com),
  • ChemSpider, Delphinidin | C15H11O7 | ChemSpider ,
  • Alappat, B.; Alappat, J. Anthocyanin Pigments: Beyond Aesthetics. Molecules2020, 25, 5500. https://doi.org/10.3390/molecules25235500
  • Nasirizadeh et al. (2016) https://doi.org/10.1016/j.jfda.2015.11.011, etc.).

You probably mean another possibility how to arrange double bonds in flavylium cation (as below), but we preferred keeping the generated figure in software ChemDraw to avoid errors by changes.

Finally, there is no carbon lacking bonds, in both conformations you can see 5 double bonds in 2 rings in total.

 We believe that the mentioned molecule delphinidine-3-glucoside structure is correct as it was generated by ChemDraw software, and we explained and supported the statement with other references. But after checking all the structures in Figure 1 we found missing double bond in the structure of delphinidin-3-rutinoside. It was corrected in manuscript.

Reviewer 2 Report

The authors addressed all comments and implemented most of the recommendations. I consider the article is complete and suitable for publication.

Author Response

The authors addressed all comments and implemented most of the recommendations. I consider the article is complete and suitable for publication.

Thank you for positive evaluation.